# Fabrication of 5 V High-Performance Solid-State Asymmetric Supercapacitor Device Based on MnO$_2$/Graphene/Ni Electrodes

**Ming-Chun Hsieh [1], Bo-Han Chen [1], Zhong-Yun Hong [1], Jue-Kai Liu [1], Pin-Cheng Huang [2] and Chao-Ming Huang [3],***

[1] Department of Materials Engineering, Kun Shan University, Tainan 710, Taiwan; michelhsieh@gmail.com (M.-C.H.); dd890615@gmail.com (B.-H.C.); st112926@gmail.com (Z.-Y.H.); k0972792033@gmail.com (J.-K.L.)
[2] Department of Environmental Engineering, Kun Shan University, Tainan 710, Taiwan; dolls12327@gmail.com
[3] Green Energy Technology Research Center and Department of Materials Engineering, Kun Shan University, Tainan 710, Taiwan
* Correspondence: charming@mail.ksu.edu.tw

**Abstract:** To reach high energy density and excellent cycle stability, an asymmetric supercapacitor device combining a high-power electric double-layer capacitor (EDLC) anode and high energy density battery-type cathode has been designed and fabricated. A binder-free strategy was used to prepare cathode by coating graphene (G) on Ni foam (Ni), then electrodepositing MnO$_2$, followed by calcination process. The potentiodynamic (PD) electrodeposition cycles of MnO$_2$ onto graphene significantly impact the electrochemical properties. Benefiting from the hierarchical structure and binder-free process of the designed 75 C/G/Ni hybrid cathode, potentiostatic (PS) electrodeposition followed by PD electrodeposition for 75 cycles demonstrates a high specific capacitance of 691 F g$^{-1}$ at 2 A g$^{-1}$. The enhanced capacitive performance can be attributed to the synergistic effect between MnO$_2$ nanosheets and graphene, in which graphene can serve as ideal support matrix and conductive channels. Furthermore, an asymmetric supercapacitor was fabricated with 75 C/G/Ni and (G + AC)/Ni as the cathode and anode, respectively, and a carboxymethyl cellulose–potassium hydroxide (CMC-KOH) gel electrolyte. The 75 C/G/Ni//(G + AC)/Ni asymmetric supercapacitor (ASC) exhibits a maximum energy density of 43 kW kg$^{-1}$ at a power density of 302 W kg$^{-1}$ with a potential window of 1.6 V and maintains good cycling stability of 88% capacitance retention at 2 A g$^{-1}$ (over 5000 cycles). Four solid-state asymmetric supercapacitors stack connected in series display an effective 5.0 V working potential to increase the voltage and output energy as a device. The device was charged using a 18,650 Li battery with a voltage of +3.8 V for 30 s and discharged six white LEDs for 20 min. The facile fabrication and remarkable capacitive performance of the MnO$_2$/G/Ni hybrid make it a promising electrode candidate in electrochemical energy conversion/storage devices.

**Keywords:** graphene; MnO$_2$; binder-free; asymmetric supercapacitor; LED





## 1. Introduction

The impacts of global warming, resulting from greenhouse gas emissions, on the environment, agriculture, and human life have been an urgent issue worldwide in recent years. One of the solutions to reduce carbon and greenhouse gas emissions is the replacement of fossil fuel vehicles with electric vehicles (EV) in the transportation sector. Lithium-ion batteries and supercapacitors comprise hybrid energy-storage systems, which possess different advantages and disadvantages to make EVs competitive with fossil fuel vehicles [1]. Lithium-ion batteries are the mainstream energy supply in EVs; however, the safety concern resulting from flammable organic electrolytes employed needs to be solved. In the case of supercapacitors, they are used to offer the high power required for short-term acceleration and restore energy during braking while the EV is operating. Supercapacitors

provide improved safety, a wide operating temperature range, and long cycling stability compared to batteries.

The main shortcoming of supercapacitors is the low energy density, which fails to satisfy the energy supply demand in advanced EVs. Overcoming this obstacle, the increase in the energy density of supercapacitors is crucial to meeting future energy demands. Raising the specific capacitance (C) and the operation voltage (V), according to the equation $E = 1/2\ CV^2$, can enhance the energy density of supercapacitor cells. The ASC, operated in a wide potential window, represents a promising approach to achieving high energy density without lowering power density, which consists of a faradic (positive) electrode and a capacitor-type (negative) electrode using appropriate electrolytes [2,3].

As far as the electrode materials are considered, they can be generalized as high-power electric double-layer capacitor (EDLC) electrodes and high-energy-density battery-type electrodes such as $Co_3O_4$, $Fe_2O_3$, $MnO_2$, and $NiO$. Manganese dioxide ($MnO_2$) is the most promising due to its low cost, toxicity, natural abundance, and high theoretical specific capacitance. Nevertheless, its poor conductivity, easy dissolution in electrolytes, and irreversible crystal volume expansion have hindered its commercial applications. To address these issues, the combination of $MnO_2$ with conductive carbonaceous materials (AC, CNT, graphene, and rGO) to form hybrid electrodes has been proposed. Graphene (G) is a two-dimensional thin-layered material with *sp*2 hybridized carbon lattice. Owing to its high surface area, remarkable chemical stability, and excellent mechanical property, graphene research has become one of the trends in developing nanosensors [4–11] and electrode materials for ASCs.

Concerning the electrolytes, aqueous electrolytes such as KOH, $H_2SO_4$, and $Na_2SO_4$ are widely applied to supercapacitors because of their low resistance, high conductivity, low cost, easy preparation, and nonflammability. The main shortcoming of the aqueous electrolyte is its narrow operation voltage window (approximately 1.2 V) due to water decomposition at 1.23 V, which limits the use of the ASCs. Lee et al. [12] used carboxymethyl cellulose (CMC) combined with a lithium nitrate salt as a gel electrolyte. They reported that coconut-shell-based carbon solid-state supercapacitors with the gel electrolyte had a potential operating window of 2.0 V and an energy density of 49 Wh kg$^{-1}$ with a 92% capacitance retention at 2 A g$^{-1}$ after cycling for 5000 times. An operating voltage of up to 2 V can be achieved with a gel electrolyte, which stops the dissolution of the active electrode material into the electrolyte, improving cycling stability.

In this work, a 3D binder-free high-energy composite electrode was prepared by facial coating, electrodeposition, and calcination process for energy storage application. Graphene (G) was coated on Ni foam (Ni), which provides a highly conductive platform for charge transport, while $MnO_2$ electrodeposited onto G/Ni produces a synergistic effect for enhanced electrochemical energy storage. The electrodeposited $MnO_2$ can avoid using any binders, which generally causes the reduction of the electrode conductivity. Besides, uniform and easy control of different mass loading and morphologies of $MnO_2$ can be obtained. Furthermore, ASC—consisting of the $MnO_2$/G/Ni composite cathode, graphene plus mesoporous activated carbon anode, and a carboxymethyl cellulose–potassium hydroxide (CMC-KOH) gel electrolyte—exhibits excellent overall electrochemical performances, including high energy density, power density, and good cycling stability. It reveals an effective protocol for high-performance solid-state supercapacitors with low-cost materials, facile and scalable fabrication toward energy storage, and beyond.

## 2. Results and Discussion

### 2.1. Raman Analysis

Figure 1 exhibits the Raman spectra of G/Ni, $MnO_2$/Ni, and $MnO_2$/G/Ni samples. As shown in Figure 1a, two prominent peaks can be observed at 1348 cm$^{-1}$ and 1586 cm$^{-1}$, and these can be assigned to the D and G bands of graphene, representing the structural defects and in-plane vibration of the sp2 carbon atoms, respectively. Moreover, the G/Ni sample displayed a relatively pronounced 2D band at ~2695 cm$^{-1}$ and a minute D + G

band at ~2930 cm$^{-1}$, implying a multilayer graphene structure. As for pristine MnO$_2$, the Raman spectrum shows several characteristic peaks at 314, 357, and 642 cm$^{-1}$, confirming the presence of α-MnO$_2$. After depositing MnO$_2$ to G/Ni, the characteristic D, G, and 2D bands of graphene were still observed, and the peak located at 642 cm$^{-1}$ is assigned to the Mn-O stretching vibration of MnO$_6$ octahedral.

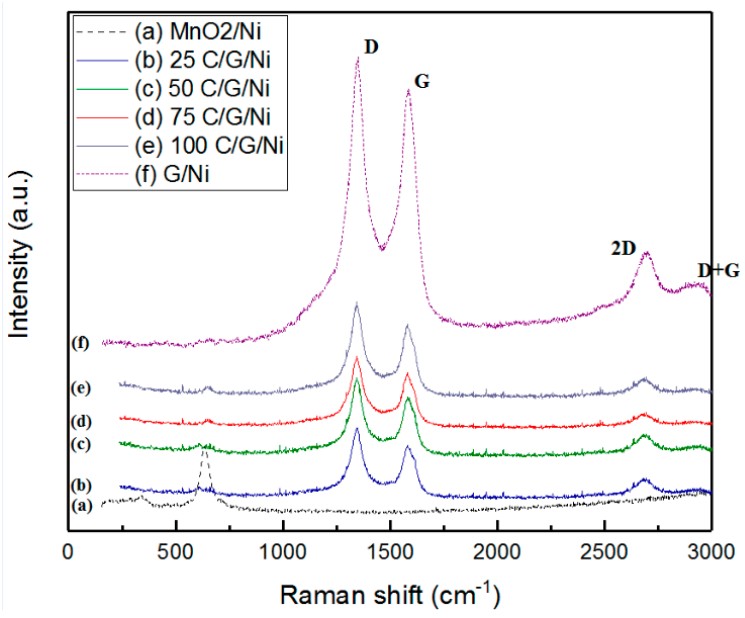

**Figure 1.** Raman spectra of (**a**) MnO$_2$/Ni, (**b**) 25 C/G/Ni, (**c**) 50 C/G/Ni, (**d**) 75 C/G/Ni, (**e**) 100 C/G/Ni, (**f**) G/Ni.

## 2.2. SEM and EDX Analysis

The morphology of the pristine MnO$_2$/Ni and MnO$_2$/G/Ni composite electrodes are illustrated in Figure 2. As shown in Figure 2a, striplike structures bound together as a 3D network of the pristine MnO$_2$ were observed over the skeleton of the Ni foam substrate. Figure 2b–e indicate the effect of the PD electrodeposition cycle on the microstructure of MnO$_2$/G/Ni composite. In the 25 cycles of PD electrodeposition, a flowerlike surface morphology with a diameter of 500~600 nm consisting of multiple petaloid MnO$_2$ nanoflakes interconnected to each other was observed for 25 C/G/Ni (Figure 2b). Increasing the cycle to 50 times, as presented in Figure 2c, drives the fusion of the flower structures to form nodular morphology with grains of approximately less than 0.3 μm in length. When the deposition cycle increased to 75, dense nanograins ranging from 40–60 nm dimensions were uniformly deposited. In contrast, for the 100 cycles, the G/Ni substrate was wholly covered with aggregated nanoflakes in a leaflike morphology (Figure 2e). The morphology of the prepared MnO$_2$/G/Ni composite samples strongly depends on the electrodeposition cycle. Though PS was followed by PD deposition, the 25 C/G/Ni-75 C/G/Ni surfaces were coated uniformly with hierarchical porous MnO$_2$ (Figure 2b–d). These as-formed MnO$_2$/graphene films had loose porous nanostructures with plenty of space. These structures can promote the mass transport of electrons for rapid redox reactions and double-layer charge/discharge processes, which is vital for effectively utilizing the electrode/electrolyte contact area during the electrochemical reaction between the MnO$_2$/G/Ni electrode and KOH electrolyte. Furthermore, the EDX spectrum of 75 C/G/Ni (Figure 2f) shows that the main elements are Mn, O, C, and Ni, and an atomic ratio of Mn and O of 1:2, further confirming the formation of the MnO$_2$ structure.

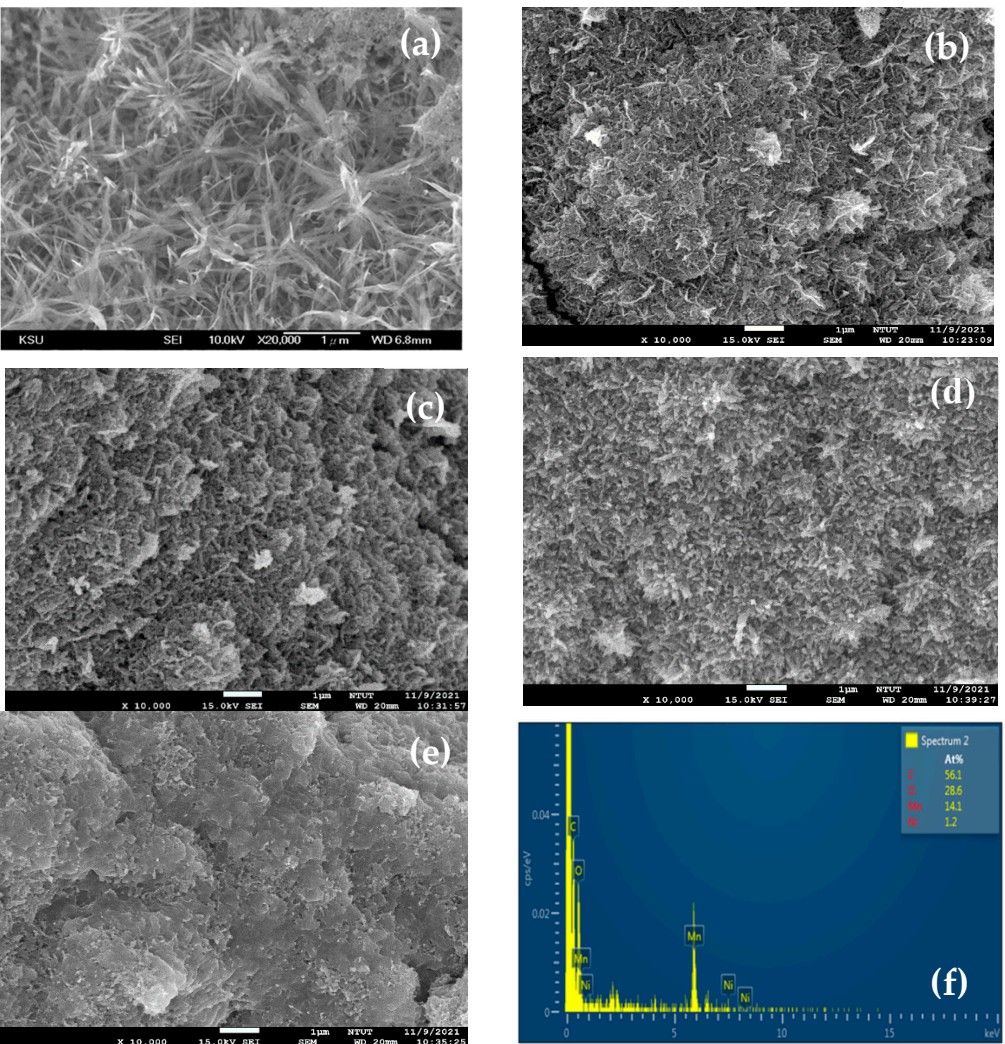

**Figure 2.** SEM images of (**a**) MnO$_2$/Ni, (**b**) 25 C/G/Ni, (**c**) 50 C/G/Ni, (**d**) 75 C/G/Ni, (**e**) 100 C/G/Ni, and (**f**) EDS spectrum of 75 C/G/Ni.

### 2.3. Electrochemical Evaluations

2.3.1. CV and GCD Analysis of Electrodes

Through CV and GCD, we investigated the effect of the cycle of PS electrodeposition on the electrochemical performances, and the results are shown in Figure 3. The CV measurements for as-prepared electrodes measured at 50 mV s$^{-1}$ rate over the potential window of 0.0–0.6 V in 4 M KOH electrolyte solution are shown in Figure 3a. All CV curves of MnO$_2$/Ni (prepared by PD electrodeposition for 75 cycles) and MnO$_2$/G/Ni (25 C–100 C) exhibited pseudocapacitive characteristics with a pair of redox peaks, which showed no noticeable distortion except a slight peak position shift because of the polarization effect of the electrode. The redox peaks originate from the reversible Faradaic reaction of Mn$^{4+}$/Mn$^{3+}$ in an alkaline electrolyte according to Equation (1), which involves intercalation/deintercalation of the electrolyte cations (K$^+$) in the bulk crystalline of MnO$_2$ [13].

$$(MnO_2)_{bulk} + K^+ + e^- \leftrightarrow (MnOOK)_{bulk} \qquad (1)$$

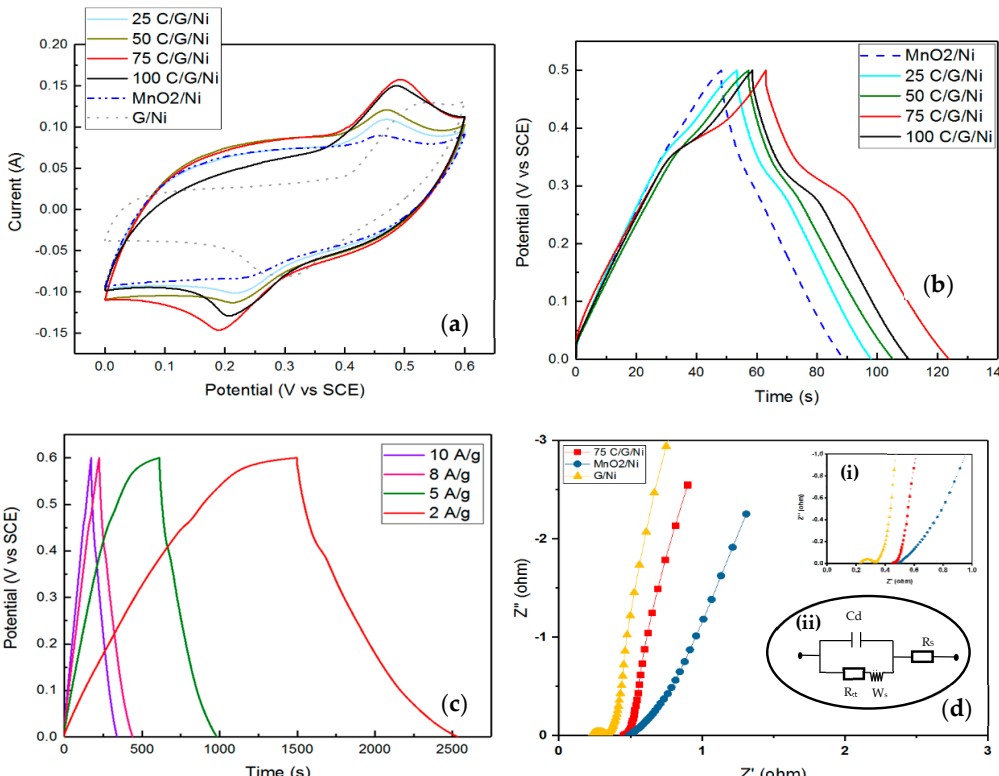

**Figure 3.** Electrochemical characteristics of as-prepared electrodes in 4 M KOH aqueous electrolyte in a three-electrode system: (**a**) CV curves of G/Ni, $MnO_2$/Ni, and $MnO_2$/G/Ni electrodes at the scan rate of 50 mV/s; (**b**) GCD curves of $MnO_2$/Ni, and $MnO_2$/G/Ni electrodes at the current density of 5 A g$^{-1}$; (**c**) 75 C/G/Ni electrode (3.5 × 7 cm$^2$) at various current densities; (**d**) Nyquist plots of G/Ni, $MnO_2$/Ni, and 75 C/G/Ni electrodes (3.5 × 7 cm$^2$); inset (i) shows an enlarged EIS spectra and inset (ii) represents the equivalent circuit diagram.

As shown in Figure 3a, it was observed that the electrochemical performance, such as redox peak current and enclosed area, of the $MnO_2$ samples supporting the graphene layer was higher than that of the $MnO_2$/Ni electrode and increased with the cycles of PD electrodeposition. The biggest enclosed area of the CV curve of 75 C/G/Ni compared with the other three $MnO_2$/G/Ni electrodes, indicating the largest specific capacitance. Figure 3b compares the GCD curves of the as-prepared $MnO_2$/Ni and $MnO_2$/G/Ni electrodes at the current density of 5 A g$^{-1}$. A little deviation from the symmetric triangular curve of pristine $MnO_2$ was observed, indicating rapid charge–discharge rates. The charge–discharge curves of all $MnO_2$/G/Ni electrodes exhibited a nonlinear shape with battery-like characteristics, revealing the noncapacitive faradaic reaction. It is noted that there is a plateau at the voltage around 0.28 V (vs. SCE) for 75 C/G/Ni and 100 C G/Ni electrodes, signifying the typical pseudocapacitance behavior to the electrochemical adsorption/desorption or redox reaction at the $MnO_2$ interface [14]. According to the GCD curves, the specific capacitances of electrodes were calculated using the equation $C = (I \times \Delta t)/(m \times \Delta V)$, where $C$ is the specific capacitance (F g$^{-1}$). $I$, $\Delta t$, $m$, and $\Delta V$ are the discharging current (mA), discharging time (s), mass loading of active materials (mg), and potential drop (V) in the galvanostatic discharge process, respectively. The order of specific capacitance for these five as-prepared $MnO_2$/Ni and $MnO_2$/G/Ni electrodes is 75 C/G/Ni (609 F g$^{-1}$) > 100 C/G/Ni (518 F g$^{-1}$) > 50 C/G/Ni (479 F g$^{-1}$) > 25 C/G/Ni (446 F g$^{-1}$) > $MnO_2$/Ni (407 F g$^{-1}$). The variation of specific capacitance from GCD shows a trend similar to that obtained from the CV curves, and the 75 C/G/Ni electrode also reveals the longest discharge time as having the highest performance. The results indicated that the presence of graphene layer in the $MnO_2$/G/Ni composite electrodes plays a key role in enhancing the capacitance; the graphene layer provides plentiful active sites for electrodeposition of $MnO_2$. Moreover, the graphene layer

serves as an excellent electrical conductor between the $MnO_2$ and Ni foam. The slight increase in specific capacitances of 25 C/G/Ni electrodes compared with the $MnO_2$/Ni electrode may be ascribed to a small amount of $MnO_2$ electrodeposited on the graphene surface (Figure 2b). As the mass loading of $MnO_2$ increased, specific capacitances increased significantly. The specific capacitance first increased, then decreased concerning $MnO_2$ loadings, and the highest value of 609 F $g^{-1}$ was obtained for 75 C/G/Ni. When the cycles increase to 100, the denser and compact structure may influence cations migrating into the $MnO_2$ layer. The capacitance of 100 C/G/Ni electrodes was decreased to 518 F $g^{-1}$, as observed in the SEM image (Figure 2e). To increase the energy storage of composite electrodes, a larger size of 3.5 × 7 $cm^2$ 75 C/G/Ni electrodes was fabricated. The GCD performance of larger size 75 C/G/Ni electrode is shown in Figure 3c at different current densities. It is evident that the charging currents at low current densities (2 A $g^{-1}$ and 5 A $g^{-1}$) were asymmetric to their discharging counterparts and revealed the noncapacitive faradaic reaction mechanism again during the charging and discharging process. The 75 C/G/Ni electrodes exhibited a specific capacity of 691, 620, 572, and 555 F $g^{-1}$ at 2, 5, 8, and 10 A $g^{-1}$. The highest specific capacitance, 620 F $g^{-1}$ at 2 A $g^{-1}$, was significantly larger than those of most $MnO_2$–graphene layer fabricated electrodes in previous studies [11–16]. EIS measurements were carried out to evaluate the combined resistance of the G/Ni, 75 C/G/Ni, and $MnO_2$/Ni electrodes. Impedance spectra of G/Ni, $MnO_2$/Ni, and 75 C/G/Ni electrodes are illustrated in Figure 3d, with an enlarged view (inset i) and a fitted equivalent circuit (inset ii). The Nyquist plot of EIS displayed a small semicircle or little semicircle in the high-frequency region and a linear line in the low-frequency region. The high-frequency intercept indicates the equivalent series resistance of the ionic resistance of electrolyte, the interface resistance of electrode/electrolyte, and the intrinsic resistance of substrate. As shown in the inset of Figure 3d, the equivalent series resistance ($Rs$) values of G/Ni, 75 C/G/Ni, and $MnO_2$/Ni were very small at 0.22, 0.42, and 0.51 Ω, respectively. The semicircle diameter along the *x*-axis in the high-to-medium-frequency region is indicative of the charge-transfer resistance ($R_{ct}$) between the electrolyte/electrode materials. According to the semicircle diameter, the $R_{ct}$ of G/Ni and 75 C/G/Ni are 0.1 Ω and 0.02 Ω. In the low-frequency region, the straight line is the Warburg resistance ($W_s$), responding to the electrode's ion diffusion resistance and capacitive behavior [17]. The more vertical the line, the lower the diffusion resistances and better the capacitor behavior. A nearly vertical line of the G/Ni electrode displayed good capacitive behavior and was closer to an ideal supercapacitor. The slope of the straight line for 75 C/G/Ni electrode was similar to that of the G/Ni, but much larger than that of the $MnO_2$/Ni electrode, indicating much lower diffusive resistance than the $MnO_2$/Ni. The 75 C/G/Ni electrode not only has a lower series resistance but also has a smaller diffusive resistance of electrolyte in the electrode than those of $MnO_2$/Ni. The high specific capacitance and low resistance of the 75 C/G/Ni electrodes may be attributed to the followed reasons: (1) A nodular morphology with a highly porous structure provides numerous robust reservoirs to enhance electrolyte penetration and shorten the diffusion distance of electrolytes to the interior surfaces of $MnO_2$ for redox reactions; (2) graphene constructs a highly intrinsic conductive network for charge transport to significantly increase electronic conductivity; (3) the synergistic effect from $MnO_2$ and graphene improves both the capacitive non-Faradaic (EDLC) and capacitive Faradaic (pseudocapacitive).

### 2.3.2. Assembly and Performance of ASC Cell

The highest performance electrode, 75 C/G/Ni, was further used as the positive electrode for the ASC. An asymmetric supercapacitor was assembled with 75 C/G/Ni and (G + AC)/Ni in 3.5 × 7 $cm^2$ as the positive and negative electrodes in the CMC-KOH gel electrolyte (Scheme 1). Before assembling the ASC cell, the stable potential window and mass ratio of the 75 C/G/Ni and (G + AC)/Ni were studied. The CV measurements of the two electrodes were studied using a three electrode system with platinum foil as counter electrode and a saturated calomel electrode (SCE) as reference electrode in 4 M

KOH aqueous solution. The 75 C/G/Ni electrode was measured within a potential window of 0 to 0.6 V (vs. SCE), while (G + AC)/Ni was measured within a potential window of –1.0 to 0 V (vs. SCE) at a scan rate of 50 mV s$^{-1}$ (Figure 4a). The CV curve of redox behavior of 75 C/G/Ni and quasi-rectangular shape of (G + AC)/Ni was observed, and the potential of the assembled ASC can be operated up to 1.6 V as the sum of the potential range of 75 C/G/Ni and (G + AC)/Ni. The specific capacity C of the half cell was evaluated from the CV curve using the relation given by Equation (2):

$$C = \frac{1}{mv(V_2 - V_1)} \int_{V_1}^{V_2} I(V)dV \tag{2}$$

where m, $v$, $V_2 - V_1$, I are mass of active material ($3.5 \times 7$ cm$^2$ area) in mg; the scan rate is in mV·s$^{-1}$, potential window in $V$, and response current in mA. The 75 C/G/Ni and (G + AC)/Ni mass ratio was obtained based on the 75 C/G/Ni and (G + AC)/Ni charge balance. The charge balance follows the relationship Q+ = Q−. The charge stored (Q) depends on the C (specific capacitance), $\Delta V$ (potential window during the charge/discharge process), and m (the mass loading of active material) following $Q = C \times \Delta V \times m$. Based on 75 C/G/Ni and (G + AC)/Ni electrodes, the optimized mass ratio of $m_{75C/G}$ to $m_{(G+AC)}$ should be 1.13 in the ASC cell. Figure 4b exhibits the CV curves of an optimized 75 C/G/Ni//(G + AC)/Ni ASC operated at various potential windows, 1.2–1.6 V, at a scan rate of 50 mV s$^{-1}$. When the voltage expanded to 1.6 V, the shape of CV still retained a quasi-rectangular shape without obvious distortion; therefore, the potential window of the as-fabricated ASC was set to 1.6 V to investigate the overall electrochemical performance further. The rate-dependent CVs of 75 C/G/Ni//(G + AC)/Ni at various scan rates from 5 to 100 mV s$^{-1}$ is represented in Figure 4c. The CV curves have no obvious changes with increasing scan rate and still kept relatively quasi-rectangular profiles, indicating the distinguished rate capability of ASC cell. Moreover, the quasi-rectangular shape of the CV curve of the ASC was also observed at a high scan rate of 100 mV s$^{-1}$, implying that the low contact resistances in the 75 C/G/Ni and (G + AC)/Ni electrodes due to graphene acted as the electronic conductive channels to increase electrical conductivity. Figure 4d depicts the charge–discharge plots of the ASC at various current densities from 2 to 20 A g$^{-1}$ in the potential range of 0–1.6 V. The charge curves are nearly symmetric with the corresponding discharge counterparts, and no obvious IR drop was observed in any of the charge–discharge curves, illustrating a rapid I–V response, small equivalent series resistance, and excellent electrochemical reversibility of the ASC. Based on the galvanostatic charge–discharge curves, the specific capacitance of the ASC was calculated, and the mass specific capacitances of the ASC at 2, 5, 8, 10, and 20 A g$^{-1}$ were 120, 68, 17, 16, and 5 F g$^{-1}$, respectively, where the total mass of the active materials in both electrodes was about 4.9 mg cm$^{-2}$, measuring 3.5 cm × 7 cm. The cycling stability of the ASC was evaluated by repeated charge–discharge tests in the potential range of 0–1.6 V (vs. SCE) at a constant current density of 2 A g$^{-1}$ between 0 and 1.6 V. The capacitance retention ratio as a function of cycle number is displayed in Figure 4e. After 5000 cycles, the capacitance remained at 88% of the initial capacitance, demonstrating excellent long-term cycling durability. The inset figure shows the GCD profiles of four solid-state ASCs stacks connected in series under repeated charge–discharge tests at 100 cycles under the current density of 2 A g$^{-1}$ in the potential range of 0–5.0 V, indicating that both electrochemical double-layer capacitance and pseudocapacitance behaviors occurred in the ASC. Figure 4f depicts the overlay of the EIS spectra at the first and 5000th cycles for the assembled device. After 5000 cycles, the equivalent series resistance slightly increased from 0.50 to 0.55 Ω. Before cycling, the high line gradient in the low-frequency region showed low interfacial diffusion resistance; however, a lower slope was observed after 5000 cycles, probably due to an increase in the charge-transfer resistance and a slower diffusion rate of ions between electrode and electrolyte after cycling.

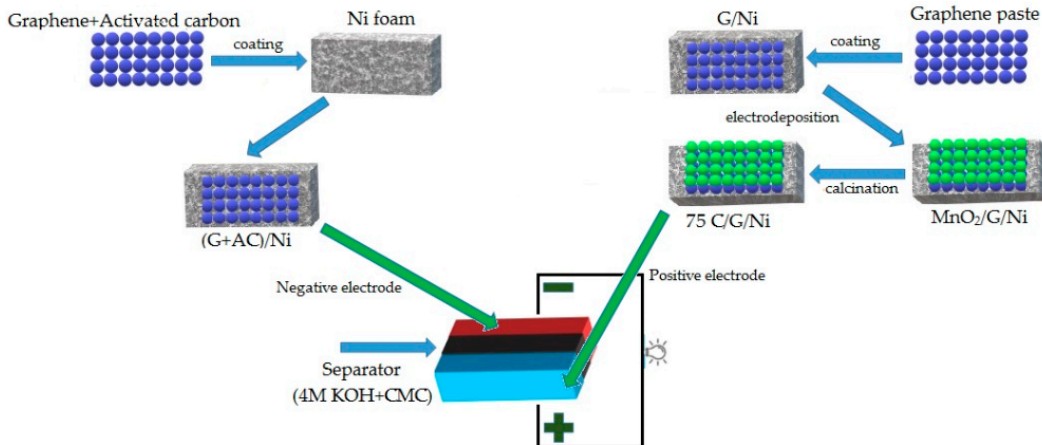

**Scheme 1.** Schematic illustration of the fabricated asymmetric supercapacitor device based on 75 C/G/Ni as positive electrode and (G + AC)/Ni as negative electrode in a CMC-KOH(4 M) gel electrolyte.

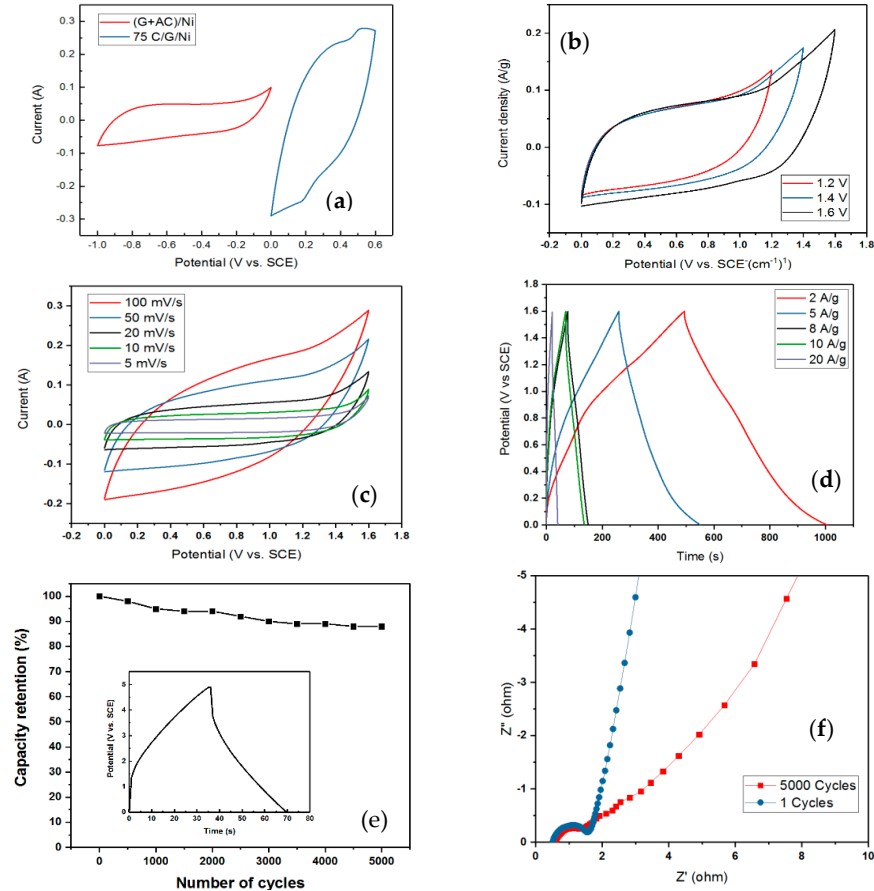

**Figure 4.** Electrochemical performance of 75 C/G/Ni//(G + AC)/Ni asymmetric supercapacitor: (**a**) comparative CV curves of 75 C/G/Ni as positive electrode and (G + AC)/Ni as negative electrode measured at a scan rate of 50 mV s$^{-1}$ in 4 M KOH electrolyte; (**b**) CV curves measured over different potential window at the scan rate of 50 mV s$^{-1}$; (**c**) CV curves measured at various scan rates; (**d**) GCD curves at various current densities; (**e**) cycling performance at a current density of 2 A g$^{-1}$ (inset: the charge–discharge curves of 100 cycles for four solid-state ASCs connected in series); (**f**) Nyquist plots of the initial and after 5000 cycles.

Energy density and power density are two key factors to evaluate the practical utility of the energy storage devices. The energy density ($E$, Wh kg$^{-1}$) and power density ($P$, W kg$^{-1}$) are calculated according to GCD curves employing the following equations:

$$E = \frac{C_S \times \Delta V^2}{2 \times 3.6} \qquad (3)$$

$$P = \frac{3600 \times E}{\Delta t} \qquad (4)$$

where $Cs$ (F g$^{-1}$) is the gravimetric specific capacitance of the ASC, $\Delta V$ (V) is voltage drop upon discharging, and $\Delta t$ (s) is the discharge time.

The energy density ($E$) and power density ($P$) of the assembled ASC device were calculated based on the GCD curves using Equations (3) and (4), and the Ragone plots of the ASC are shown in Figure 5. It is notable that 75 C/G/Ni//(G + AC)/Ni ASC delivered a maximum energy density of 43 Wh kg$^{-1}$ at the power density of 302 W kg$^{-1}$ with a voltage window of 1.6 V. The electrochemical performance was comparable or higher than the MnO$_2$–graphene-based asymmetric supercapacitors reported previously, such as MnO$_2$–graphene//graphene (91 Wh kg$^{-1}$ at a power density of 400 W kg$^{-1}$) [16–19], MnO$_2$–graphene–carbon nanofiber//graphene–carbon nanofiber (23 Wh kg$^{-1}$ a power density of 451 W kg$^{-1}$) [20], MnO$_2$–graphene–carbon nanotube//graphene–carbon nanotube (35 Wh kg$^{-1}$ at a power density of 426 W kg$^{-1}$) [21], MnO$_2$–graphene//CNT–graphene (32 Wh kg$^{-1}$ at a power density of 454 W kg$^{-1}$) [22], MnO$_2$–graphene//MnO$_2$ (22 Wh kg$^{-1}$) [23], and MnO$_2$ nanotube–MnO$_2$ nanoflake–graphene//activated carbon (23 Wh kg$^{-1}$ at a power density of 120 W kg$^{-1}$) [24].

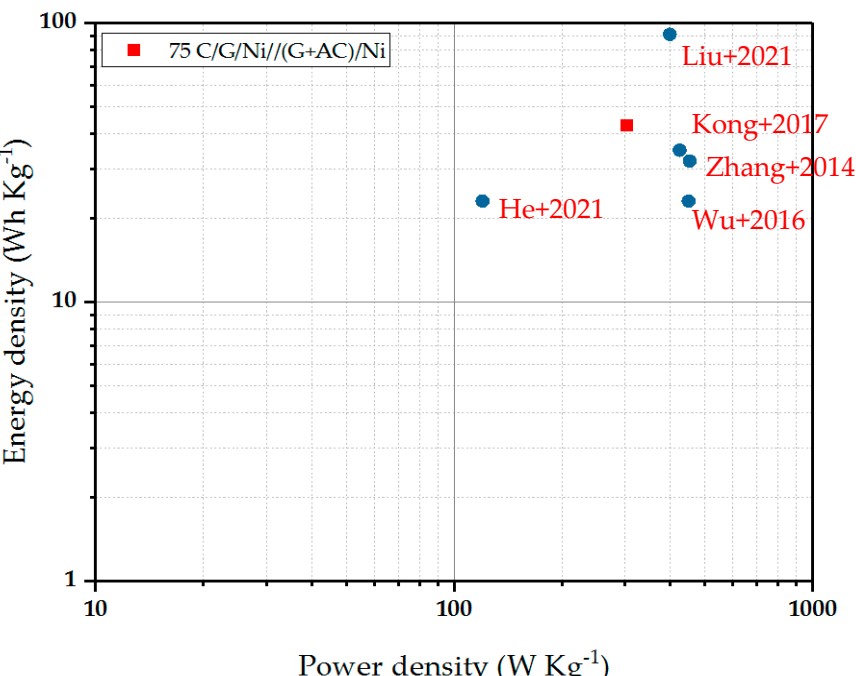

**Figure 5.** Ragone plots of 75 C/G/Ni//(G + AC)/Ni asymmetric supercapacitor compared with data in other literature [16,20–22,24].

Four solid-state 75 C/G/Ni//(G + AC)/Ni ASCs (each with an area of 64 cm$^2$) were welded in series to increase the voltage and output energy. The device was charged using a 18,650 Li battery with a voltage of +3.8 V for 30 s. Figure 6 shows a bulb of six white light-emitting diodes (LEDs) that can be lit by the device for 20 min, demonstrating the superior performance and practical application as an energy storage pack. The superior electrochemical performances of the 75 C/G/Ni//(G + AC)/Ni ASCs result from three

facets. Firstly, the binder-free process could increase the adhesion of $MnO_2$ and graphene to the bare conducting Ni foam substrate and effectively reduce the electrical resistance between the active materials with the Ni foam substrate. Secondly, the graphene layer can serve as a continuous conductive network for electron transport and support the substrate for forming hierarchical porous $MnO_2$, which provides abundant electroactive sites to increase the contact area between electrolyte with $MnO_2$ for pseudo-capacitor reaction. Thirdly, calcination improves the interfacial contact force between $MnO_2$ and graphene, maintaining the structure's integrity during the long cycle process.

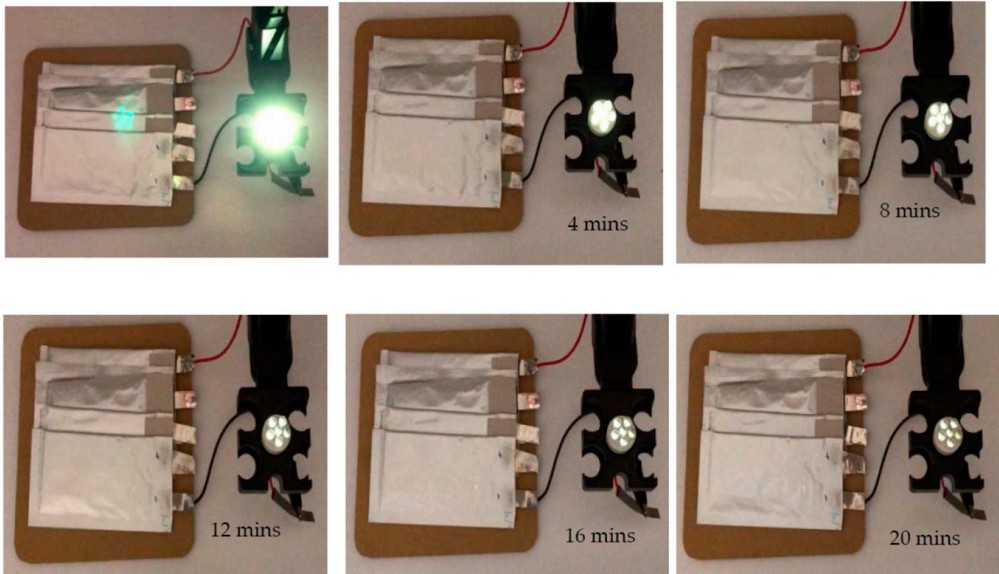

**Figure 6.** Four ASCs attached in series light up a bulb for 20 min, which consists of 6 white LEDs.

Recently, several reports have discussed graphene–$MnO_2$ powder catalysts as visible light active photocatalysts to degrade industrial and pharmaceutical pollutants [25–27]. In this study, the above results exhibit that 3D binder-free $MnO_2$–graphene composite prepared by a facial coating, electrodeposition, and calcination process is a promising design for energy storage applications due to its simple, cost-effective, and facile scale-up preparation method with high specific electrochemical performance.

## 3. Experimental Section

### 3.1. Material Details

Manganous acetate (Sigma, St. Louis, MO, USA, >99%) and sodium sulfate (Sigma, St. Louis, MO, USA, >99%) were used as received for the preparation of $MnO_2$. The graphene pastes (TCMC, Taoyuan, Taiwan, E-Closer® 040 WB, 4% active content) were used without further purification. Nickel foam (UBIQ, Taoyuan, Taiwan, 110 pores per square inch) was used as received as the substrate of electrode. Commercially available mesoporous activated carbon (CSCC, Taiwan, ACS25, BET $2500 \pm 200$ $m^2/g$) was used without further treatment.

### 3.2. Preparation of $MnO_2$/G/Ni Electrodes and (G + AC)/Ni Electrodes

Before electrode preparation, Ni foams were cleaned through acetone ultrasonically in deionized (DI) water for 5 min and dried at 100 °C under vacuum for 2 h. $MnO_2$/G/Ni was obtained by a facile two-step method. The dilute graphene paste was directly coated uniformly onto the Ni foam and dried in a vacuum oven (100 °C, 2 h) for a mass loading of graphene ~1 mg. Subsequently, $MnO_2$ was grown in situ on the G/Ni with electrodeposition performed in the electrochemical station (6273E, CH Instruments Ins., Austin, TX, USA). Briefly, a three-electrode system was used with a Pt plate as a counter electrode, a saturated calomel reference electrode (SCE), and a piece of G/Ni ($2 \times 3$ $cm^2$) as the

working electrode. As electrolyte, 0.1 M $Mn(CH_3COO)_2 \cdot 4H_2O$ and 0.1 M $Na_2SO_4$ was used. The electrodeposition was conducted with two modes, i.e., PD and PS electrodepositions. The PS electrodeposition of $MnO_2$ was carried out by applying voltage + 0.6 V/SCE for 900 s, followed by PD electrodeposition between the potential limits of +0.3 to +0.6 V/SCE at 25 mV s$^{-1}$ scan rate for 25, 50, 75, and 100 cycles, named as 25 C/G/Ni, 50 C/G/Ni, 75 C/G/Ni, and 100 C/G/Ni, respectively. After electrodeposition, the samples were thoroughly rinsed with DI water, dried, then annealed at 300 °C for 2 h in the air. The masses of the electrodeposited $MnO_2$ of $MnO_2$/G/Ni samples were 1.10 (±0.01), 1.23 (±0.02), 1.31 (±0.02), and 1.39 (±0.03) mg cm$^{-2}$, respectively. For comparison experiments, graphene paste directly coated on the Ni foam with oven-dry and $MnO_2$ (PD electrodeposition between the potential limits of +0.3 to +0.6 V/SCE at 25 mV s$^{-1}$ scan rate for 75 cycles) electrodeposited on the Ni foam without graphene layer were named G/Ni and $MnO_2$/Ni, respectively, with 1.23 (±0.03) and 1.25 (±0.03) mg cm$^{-2}$.

(G + AC)/Ni was obtained by a coating method. The graphene paste was dispersed in water under stirring as a graphene dispersion with the concentration of 2.0 mg mL$^{-1}$. Then, a certain amount of AC (ACS25) was added to graphene dispersion under continuous stirring. The as-prepared mixed paste was coated uniformly on the precleaned Ni foam without any adhesives or any conductive agents. Further, the coated sample was dried, measured, and weighed after each coating. The coating step was repeated to achieve uniform film with graphene 1.25 (±0.02) and AC 0.2 (±0.01) mg cm$^{-2}$, respectively.

### 3.3. Sample Characterization and Electrochemical Measurements

The surface morphologies and elemental compositions were examined by scanning electron microscopy (SEM, JEOL JSM-6700F, Tokyo, Japan) equipped with energy-dispersive X-ray spectroscopy (EDS). The structures were investigated by Raman spectrometer (Raman, UniDRON, New Taipei City, Taiwan) fitted with an Ar laser with a 532 nm wavelength. The electrochemical measurements, including cyclic voltammetry (CV), galvanostatic charge/discharge (GCD), and electrochemical impedance spectroscopy (EIS), were carried out using an electrochemical workstation (CHI 6273E, Austin, TX, USA). The specific capacitance $C$, according to the discharge curve of the GCD measurement, was obtained as $C = [(I \times \Delta t)/(\Delta V \times m)]$, where $i$ (A) is the discharging current, $\Delta t$ (s) is the discharging time, $\Delta V$ (V) is the discharging potential difference, and $m$ is the mass of the loaded active materials. EIS measurements were executed using an AC voltage with a 5 mV amplitude in a frequency range of $10^{-2}$ to $10^5$ Hz at open circuit potential.

### 4. Conclusions

A $MnO_2$/G/Ni composite as a binder-free supercapacitor electrode has been fabricated by a facial coating of graphene and controllable electrodeposition of $MnO_2$. Electrochemical measurements reveal that the 75 C/G/Ni electrode—PS electrodeposition followed by PD electrodeposition for 75 cycles—exhibits much higher specific capacitance and lower equivalent series resistance than graphene and $MnO_2$ electrodes. The synergetic combination of pseudo-capacitive material ($MnO_2$) and EDLC material (graphene) on Ni foam allows fast electron and ion transport, resulting in high electrochemical performance. Furthermore, the solid-state ASC cell, 75 C/G/Ni//(G + AC)/Ni, can deliver high specific energy of 43 kW kg$^{-1}$ at the specific power of 302 W kg$^{-1}$ and outstanding cycling stability (88% after 5000 cycles) in a potential window of 0 to 1.6 V. After being charged at +3.8 V for 30 s, the in-series four ASC cells can light up six white LEDs for 20 min. This work may provide an excellent protocol for synthesizing ASC materials based on $MnO_2$ onto graphene in scalable energy storage applications.

**Author Contributions:** M.-C.H. analysis of data and writing the first draft of the manuscript. B.-H.C.: contributed to the synthesis, experiments. Z.-Y.H. contributed to analysis of data. J.-K.L. contributed to the synthesis, experiments. P.-C.H. contributed to data interpretation. C.-M.H. conceived and designed the work and revising the manuscript. All authors have read and agreed to the published version of the manuscript.

**Funding:** This work was financially supported by the Green Energy Technology Research Center from the Featured Areas Research Center Program within the framework of the Higher Education Sprout Project.

**Data Availability Statement:** Not available.

**Conflicts of Interest:** The authors declare no conflict of interest.

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
