# Peer review of "Fabrication of 5 V High-Performance Solid-State Asymmetric Supercapacitor Device Based on MnO2/Graphene/Ni Electrodes"

_catalysts, doi:10.3390/catal12050572_

Round 1

Reviewer 1 Report

The paper is related about Fabrication of 5 V high-performance solid-state asymmetric supercapacitordevice based on MnO2/graphene/Ni electrodes. I think that the following minor points may be addressed to make the paper more readable.

- In the abstract, please add the several sentences to describe significance of the solid-state asymmetric supercapacitor.

- The English language through article should be carefully polished.

- The section of Introduction is poor. The authors used reduced graphene oxide which is preferred due to its good electrical conductivity, chemical stability with high surface area. But, the importance of these novel nanomaterials (graphene oxide/graphene quantum dots/reduced graphene oxide) and their hybrids were not indicated in manuscript. Especially, the section of introduction is very important to indicate these properties. Hence, graphene oxide/graphene quantum dots/reduced graphene oxide and its application properties such as nanosensor should be highligted in manuscript. These novel references must be included and highlighted:

1) Materials Today Chemistry, 23 (2022) 100666

2) Analytica Chimica Acta, 1200 (2022) 339609

3) New Journal of Chemistry, 45 (2021) 11222

Reviewer 2 Report

Comments: This work reported a 3D binder-free high-energy composite electrode  prepared by a facial coating, electrodeposition, and calcination progress. The as-fabricated hybrid assembled an asymmetric supercapacitor shows a maximum energy density of 43 kW kg-1 at a power density of 302 W kg-1 with a potential window of 1.6 V and maintains good cycling stability of 88 % capacitance retention at 2 A g-1 (over 5000 cycles). However, there are still some issue to be addressed before possible publication. Specific comments are given below: 

  1. In Figure 4, some numbers in the abscissa in Figure B were missing.

  1. The voltage test range of the cyclic voltammetry curve in Figure 4 seems to be 0-0.6v. Generally speaking, it should be 0-1Vfor aqueous electrolyte and higher for non-aqueous electrolyte. To the best of our knowledge, The larger the potential window, the higher the energy density. In addition, the potential windows in the charge discharge curves in Fig. B and Fig. C and the potential windows of the cyclic voltammetry curve seem to be inconsistent. What’s the reason for your choice?

  1. What is the rateperformance of the assembled capacitor under different current densities? If possible, please supplement this data.

  1. For introduction, some recently published papers onsensors maybe enriched in the revision, such as Rare Met. 40, 3520-3530 (2021)ï¼›org/10.1016/j.jmst.2021.04.054; doi.org/10.1016/j.ijhydene.2021.10.168; Industrial Crops and Products 178 (2022) 114565; Rare Met. 41, 960-971 (2022).

Reviewer 3 Report

Recommendation: Publish after minor revision.

This manuscript by the Huang group provides insights into the development of high-performance solid-state asymmetric supercapacitor devices. This work is quite interesting. However, some issues need to be addressed in the manuscript. Hence, I will recommend this manuscript be published in Catalyst only after minor revisions.

My comments to authors:

  1. The standard of English in the manuscript needs to be improved. The typos and grammatical errors should be corrected in the entire manuscript.
  2. The superscript used in the units is not consistent in the manuscript.
  3. The quality of the figure should be improved. In the figures, the colour representation of the sample should be kept constant to make the reader easier to understand. For example Fig 3(a) the 75C/G/Ni is red and in the other figure 3(b) it is blue. The size of X-Y axis labels should be increased.
  4. The full abbreviation of PD and PS should be defined in the abstract to make the reader easy to understand.
  5. What are other kinds of gel electrolytes reported in the literature apart from CMC-Lithium nitrate salt? What is the motivation for selecting the CMC-KOH?
  6. How to compare CMC-KOH with PVA with H3PO4/H2SO4/KCl/KOH type gel electrolyte?
  7. Did the author try to increase the mass loading of graphene? The authors can provide some information.
  8. It is generally known that when a composite electrode (AC/Graphene) is prepared the amount of graphene should be less and the amount of AC should be more in other words graphene should be used as an additive, not as major electrode material. (For Ex: Polymers202012(4), 765). Here G+AC is prepared in the proportion of 1.25+0.2 mg/cm2. Can the author explain what is the reason behind this proportion?
  9. How to compare the performance of 75C/G/Ni with the reported literature based on MnO2 type electrodes?
  10. The authors should provide the fitting circuit diagram in figure 3(d).
  11. Did the author try to go beyond 1.6 V for 75 C/G/Ni//(G+AC)/Ni ASC, like 1.8 or 2.0 V? Did they find any unusual distortion, or still the quasi-rectangular shape is maintained?
  12. The author should more detailed explanation of how calcination improves the interfacial contact resistance.
  13. How can the preparation involving coating, electrodeposition, and calcination is simple and cost-effective?
  14. The author should change the calcination progress → calcination process
  15. I suggest the author include a good schematic for the electrode material synthesis and electrode fabrication to improve the quality of the manuscript.

Reviewer 4 Report

In this work, the authors successfully synthesized the MnO2/graphene/Ni electrodes with maximum specific capacitance of up to a high specific capacitance of 691 F g-1. The 75 C/G/Ni//(G+AC)/Ni asymmetric supercapacitor exhibits a maximum energy density of 43 kW kg-1 at a power density of 302 W kg-1 with a potential window of 1.6 V and maintains good cycling stability of 88 % capacitance retention at 2 A g-1.

However, a few serious issues should be addressed. The detailed comments are as below,

  1. Figure 2f is not clear.
  2. Unified coordinates, the number of significant digits of coordinates are not unified in Figure 4c and b. Please check the full text
  3. “After 5000 cycles, the equivalent series resistance slightly increased from 0.50 to 0.55 Ω.” Please provide Equivalent circuit diagram provided.
  4. “When the cycles increase to 100, the denser and compact structure may influence cations migrating into the MnO2 layer.” Why?
  5. How to choose the potential window, is 1.8V OK?
